# GRAY-BOX FINE-TUNING FOR SINGLE BACKBONE DOMAIN EXPERTS

## ABSTRACT

The emergence of foundational models has greatly improved performance across various downstream tasks, with fine-tuning often yielding even better results. However, existing fine-tuning approaches typically require access to model weights and layers, leading to challenges such as managing multiple model copies or inference pipelines, inefficiencies in edge device optimization, and concerns over proprietary rights, privacy, and exposure to unsafe model variants. In this paper, we address these challenges by exploring "Gray-box" fine-tuning approaches, where the model's architecture and weights remain hidden, allowing only gradient propagation. We introduce a novel yet simple and effective framework that adapts to new tasks using two lightweight learnable modules at the model's input and output. Additionally, we present a less restrictive variant that offers more entry points into the model, balancing performance with model exposure. We evaluate our approaches across several backbones on benchmarks for text-image alignment, text-video alignment, and sketch-image alignment. Our results demonstrate that, despite having limited access to the model, our Gray-box approaches achieve competitive performance with full-access fine-tuning methods.

## 1 INTRODUCTION

The recent surge in the development of foundation models (Radford et al., 2021; Li et al., 2022; 2023; Oquab et al., 2023; Kirillov et al., 2023) has significantly advanced a wide range of downstream tasks, achieving state-of-the-art (SoTA) performance across various domains. These models are typically deployed as pre-trained backbones that have been fine-tuned to adapt them to specific domains or tasks. Common fine-tuning approaches include: 1) Full fine-tuning (Devlin et al., 2019; Dosovitskiy et al., 2021), where all model parameters are updated; 2) Partial tuning, which adjusts only a subset of parameters, often in the model's final layers (Girshick et al., 2014; Dosovitskiy et al., 2021); and 3) Integrating adapter modules (Rebuffi et al., 2017; Hu et al., 2022) into the model's layers. However, adapting large foundation models for multiple diverse sub-tasks through these conventional methods introduces several significant limitations:

1. **Duplication of deployment and storage:** Current large foundation models are costly to share and serve, and deploying a dedicated fine-tuned version for each downstream task exacerbates this burden. Managing multiple models not only increases storage and deployment complexity but also reduces efficiency, as demonstrated in the context of LLMs (Pope et al., 2023; Lester et al., 2021).

2. **Optimization for edge devices:** Adapting foundation models for deployment on edge devices requires careful optimization based on their weights and architecture (Lazarevich et al., 2021; Kwon et al., 2022). Fine-tuning models by modifying their parameters often demands repeated optimization for each device, making the process resource-intensive and inefficient, particularly for large-scale deployments.

3. **Privacy, safety, and intellectual property (IP) concerns:** Granting full access to a model's layers and weights raises risks related to IP protection, safety, and privacy. Publishing or exposing weights can lead to unauthorized use (OpenAI, 2023), or the recovery of sensitive training data (Haim et al., 2022). Moreover, it has been shown (Horwitz et al., 2024) that LoRA (Hu et al., 2022) fine-tuned models can be vulnerable to attacks capable of reconstructing the original model's weights and performance.

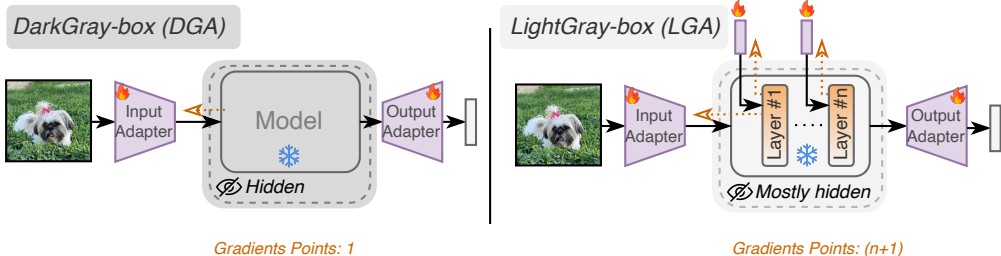

Figure 1: An overview of our gray-box frameworks. **Left:** DarkGray-Box Input/Output Adapters (DGA) permits modifications only at the input and output levels while keeping the backbone model hidden and frozen. The only information available is the gradient flow (indicated by the orange-dotted arrow), which matches the shape of the last layer of the input adapter. **Right:** In contrast, LighGray-box (LGA) allows additional entry points into the model's intermediate layers, exposing slightly more information, such as the input dimensionality and the gradients of a subset of the layers.

In this paper, we address these challenges by introducing a family of *Gray-box* fine-tuning techniques that keep the foundation model's weights and layers fixed and hidden. Conventional *White-box* techniques allow full access to the pre-trained backbone architecture and weights, but are inherently limited by the challenges mentioned earlier. In contrast, *Black-box* methods restrict access to only the model's input and output, resulting in significant performance constraints. The Gray-box approach offers a middle ground, exposing limited information about the model, which enables it to effectively address these challenges. Specifically, we consider a scenario where the provider of the backbone model offers one or more entry points to the pre-trained model (*e.g.*, the original input entry or intermediate layer entries). While keeping the weights and layers hidden, each entry point reveals: (1) the dimensionality of the layer at that entry point, and (2) the gradients of the (application-dependent) loss with respect to the entry point inputs. This Gray-box setup has practical applications in real-world scenarios. For example, in hospital models used for medical image analysis, where patient data privacy and regulatory compliance are critical, the model owner might want to allow third parties to adapt the model for specific diagnoses without exposing sensitive data or the model's proprietary structure (Bharati et al., 2022). While Federated Learning (FL) also prioritizes privacy, it primarily focuses on data privacy by distributing training across nodes. In contrast, our Gray-box framework emphasizes secure task adaptation while keeping the foundational model's structure hidden. Similarly, in persona-based models used for personalization tasks (*e.g.* personalized recommendations or identity verification), fine-tuning may be required without revealing personal data or the full model architecture (Zheng et al., 2016). Additionally, foundation model providers may wish to offer adaptation capabilities to third parties while keeping the core model architecture and weights concealed to protect intellectual property and prevent misuse. This approach allows adaptation for specific domains or tasks while minimizing the risks associated with full model exposure.

We explore two variants of the Gray-box framework: one that permits multiple entry points (thus exposing more model information) and another that restricts access to only the original input entry. We refer to these variants as *LightGray-box* and *DarkGray-box*, respectively, where the shade reflects the level of information exposed to the user during fine-tuning. Figure 1 demonstrates these settings, which offer flexible, efficient, and more secure solutions to the challenges outlined above by leveraging a pre-trained foundation model while keeping it fixed and concealed. In the following sections, we detail how our framework effectively addresses real-world challenges, enabling model adaptation with minimal exposure or modification, and demonstrating the practicality of our Gray-box approaches. A common Black-box approach to adapting an existing foundation model involves training additional layers on top of its output features (Radford et al., 2021; Devlin et al., 2019; He et al., 2022; Oquab et al., 2023). However, this method relies solely on the information provided by the model's output features, missing the opportunity to leverage the foundation model's computational power for further adaptation. Our Gray-box framework addresses this limitation by allowing modifications to the input or the injection of "middleware" features, as discussed in this paper, thereby unlocking more effective fine-tuning potential. Table 1 summarizes the benefits and requirements of these fine-tuning approaches.

Table 1: Comparison of different *"shades"* of fine-tuning methods. Each approach conceals different pieces of information regarding the backbone model and has varying requirements. The ✓ symbol indicates partial requirements or information that may vary depending on usage and often involves trade-offs. For instance, while LoRA may not require multiple backbone copies, it leads to multiple computational flows during inference. Although the zero-shot Black-box approach benefits from the most ✔ marks, DGA significantly improves zero-shot results by exposing only the gradient flow within the model.

| Approach | Hidden Information | | | Requirements | | |
|---|---|---|---|---|---|---|
| | Gradients Flow | Backbone Weights | Layers Sizes | No Layer Choice | Single Backbone Copy | Single Flow Computation |
| ☐ Full Finetune | ✗ | ✗ | ✗ | ✗ | ✗ | ✗ |
| ☐ LoRA | ✗ | ✔ | ✗ | ✗ | ✓ | ✗ |
| ▢ LGA (ours) | ✗ | ✔ | ✓ | ✗ | ✓ | ✓ |
| ▨ DGA (ours) | ✗ | ✔ | ✔ | ✔ | ✔ | ✔ |
| ■ Original (zero-shot) | ✔ | ✔ | ✔ | ✔ | ✔ | ✔ |

As evaluation, we compare our methods to four main fine-tuning alternatives: 1) Full fine-tuning, 2) Last Layers fine-tuning, 3) Light weight LoRA (Hu et al., 2022) adapter, and 4) Black-box Linear Probing. Note that the first two approaches require access to part or all of the original weights and are thus classified as white-box methods. We evaluate our methods across various tasks, domains, and backbones. We consider LoRA and full fine-tuning as the performance upper bounds in our comparisons, given their ability to modify the original model parameters. Despite this, our DarkGray-Box Input/Output Adapters (DGA) approach achieves results that are on par with, or competitive to LoRA across several tasks such as Text-to-Image Retrieval and Text-to-Video Retrieval benchmarks (*e.g.*, MSR-VTT (Xu et al., 2016), VATEX (Wang et al., 2019b), COCO (Lin et al., 2014), Flickr30K (Young et al., 2014)). We further evaluate our approach on tasks and domains that are more distant from the original backbone's focus, such as Sketch-to-Image Retrieval (Sangkloy et al., 2016) and Image/Sketch Classification (Russakovsky et al., 2014; Wang et al., 2019a).

We summarize our contributions as follows:

- We introduce a new paradigm for effectively re-use pre-trained foundation models, enabling their adaptation to new domains and tasks while balancing effectiveness, proprietary protection, safety, and efficiency, exploring various options along this spectrum.

- We propose two Gray-box frameworks, DGA and LGA, which leverage a pre-trained foundation model while keeping it intact and frozen, allowing only limited access. Our novel DarkGray-Box Input/Output Adapters (DGA) framework adapts the foundation model for new domain-specific tasks by modifying only its input and output spaces, which was not explored enough in the visual domain.

- We conduct an extensive ablation study to assess the capabilities of input and output adapters, both individually and in combination, providing deeper insights into their roles and effectiveness.

- We demonstrate the effectiveness of our Gray-box approaches across various computer vision tasks and benchmarks, achieving results that are competitive with, or on par with, White-box baselines, all while keeping the foundation model sealed.

## 2 RELATED WORK

**Prefix and Prompt Tuning** (Lester et al., 2021; Liu et al., 2021; Li & Liang, 2021) are methods proposed as lightweight alternatives to full fine-tuning for Large Language Models (LLMs). Instead of modifying all model parameters, these methods optimize a new set of input tokens for each NLP task. Prompt Tuning (Lester et al., 2021) focuses on optimizing a token sequence added to the first transformer's layer, while Prefix Tuning (Li & Liang, 2021) and Prompt Tuning 2 (Liu et al., 2021) propose optimizing a separate sequence added to each transformer layer. Due to unstable optimization when directly training prefix tokens, the Prefix-Tuning approach (Li & Liang, 2021)

trains a matrix $P$, which is projected through a trainable MLP layer to compute the prefix added to the existing prompt input. Prefix-Tuning involves learning separate prefixes for both the encoder and decoder components of the LLM, inserted appropriately during inference. Depending on the task, these methods have proven effective with prefixes ranging from 10 to 200 learned tokens, along with their associated MLP layer. In this work, we simplify this approach by directly optimizing just two tokens for a single text encoder without additional components. Specifically, we use the first token as an attached prefix and the second as a "shift" token added to all original input tokens. Consequently, our approach requires only one extra token per prompt or task, which is particularly valuable for text encoders with limited context length (*e.g.* CLIP, which is limited to 77 tokens in total).

**Low-Rank Adaptation (LoRA)** (Hu et al., 2022) was initially proposed as an effective lightweight alternative to full fine-tuning for transformer-based large language models (LLMs), and later to Vision Transformers (Dosovitskiy et al., 2021; Zhu et al., 2023). Instead of updating all model parameters, LoRA learns two $n \times r$ matrices that are multiplied to form an $n \times n$ matrix of a low rank $r$, where $r$ is a hyper-parameter. The low-rank matrix is then added to the original model's matrix. LoRA has demonstrated competitive results with full fine-tuning while being significantly more parameter-efficient. However, LoRA requires prior knowledge of the model's architecture to choose the appropriate layers, match exact dimensions, and determine the matrix ranks. For example, in transformer layers, the $Q$, $K$, and $V$ matrices across multiple layers have been shown to be effective choices for applying LoRA. Additionally, if the learned components are stored separately from the original model, LoRA necessitates a different computational flow in inference, altering the intermediate features by applying these new components. Although LoRA could be considered a "gray-box" approach due to the ability to hide the original model's weights, recent work (Horwitz et al., 2024) demonstrated methods to effectively reconstruct the original model's weights using LoRA fine-tuned models, making it more accurately associated with a "white-box" framework. Furthermore, LoRA requires custom implementations for different architectures, which have been developed for a variety of structures (*e.g.*, linear, Conv2D, embeddings). In contrast, DGA assumes no access to the model weights, no prior knowledge of internal layers, and does not require selecting any hyper-parameters. Our approach relies solely on the gradient flow through the original model and preserves the model's original structure, maintaining the inference pipeline intact between the input and output across all tasks and domains.

**Co-CoOp and MaPLe** A different lightweight fine-tuning approach is Co-CoOp (Zhou et al., 2022), a CLIP-based architecture designed to enhance the integration of visual and textual modalities for image classification. Co-CoOp concatenates the visual encoder with the textual encoder, inserting a learned network between them. It processes the image feature vector through a learned MLP, generating a fix number of visual tokens that are added as a prefix to the textual input of the text encoder. *i.e.* this approach conditions the textual input in the visual output. Although Co-CoOp keeps CLIP frozen, this design requires both modalities during each inference, limiting the generation of non-conditioned textual feature vectors, an essential capability for tasks like Image Retrieval where query (text) and images (gallery) are encoded separately. Similarly, MaPLe (Khattak et al., 2023) further improves upon Co-CoOp by learning shared vectors projected into different layers of the CLIP textual and visual encoders, using learnable MLP network. MaPLe can be seen as an extension of Prefix-Tuning (Li & Liang, 2021) for classification tasks, freezing the model and allowing internal tokens to be learned, which respects the "LightGray-box" framework. We adapt a different version of this approach to our new tasks, where indepedent vectors are learned for each layer with no shared layers that significantly increase the number of learned parameters. We refer to this light-weight approach as LGA, in this paper.

**Model thievery** has been extensively studied in the context of machine learning models (Tramèr et al., 2016; Krishna et al., 2020), particularly neural networks. Sha et al. (2023) introduced a learning approach to replicate a pre-trained transformer encoder by constructing a similar-performing encoder based on the original model's output features. Milli et al. (2019) presented techniques for reconstructing model weights, given the specific architecture of a two-layer MLP and the propagated gradients. Similarly, Horwitz et al. (2024) successfully recovered original transformer weights from LoRA fine-tuned versions of the model, while Béguelin et al. (2021) proposed a method to recover the weights of a (private) linear classification head using its (public) backbone feature extractor.

In this context, the potential theft of model weights not only poses a risk of model misuse (Bommasani et al., 2021), but also raises further concerns, as Haim et al. (2022) demonstrated a method

for recovering training data samples from the model's weights. In this work, we propose a fine-tuning framework that minimizes the risk of exposing model weights, aligning with the findings of current research. Importantly, while it is not yet practical or feasible to recover an arbitrary model's architecture and weights based solely on input gradients, we do not claim that this is impossible or inherently difficult, leaving this exploration for future research.

In summary, "White-box" and "LightGray-box" methods have been explored in NLP and classification tasks by incorporating additional components or tokens into the model's intermediate layers. While input adapters have been studied in the context of LLMs, their application in the image domain has not been thoroughly investigated, as we do in this paper. We extend this exploration through our LGA approach, which draws inspiration from these methods, and further develop a more restrictive DGA approach that preserves the original foundation model's computational flow.

## 3 METHOD

In this section, we introduce our approach for fine-tuning a foundation model $F$ (*e.g.* CLIP, BLIP) for new domain-specific tasks without exposing its architecture or modifying its weights. We propose two fine-tuning settings, termed *DarkGray-box* and *LightGray-box* settings, both of which offer lightweight fine-tuning options, and leverage the pre-trained backbone model $F$ while preserving its integrity and privacy.

### 3.1 GRAY-BOX SETTINGS

**DarkGray-box:** In this setting, the internals of $F$ are completely hidden, akin to a black-box approach. The only exposed components are the *input* and *output adapters*, which are external trainable modules plugged into the input and output of the backbone model. To train the input adapter, this setting requires access to the gradients computed by back-propagation through the backbone model. This means that a gradient tensor corresponding to the final layer of the input adapter is exposed — hence the term *DarkGray* instead of *Black*. Importantly, the backbone model's architecture and weights remain hidden, and only the adapters are trained. Our approach learns only a minimal number of parameters (approximately $0.4\%$ of the total model parameters). In this context, we address two types of input modalities: images and text.

**LightGray-box:** In this more relaxed setting, the provider introduces additional entry points where task-dependent information can be injected into the model's intermediate layers. This enables better adaptation to a domain-specific task, enhancing flexibility without compromising the advantages of the gray-box model setup. Specifically, we optimize a set of learnable tokens injected into the transformer layers of $F$, thereby influencing attention scores without accessing or modifying the weights or layers. Although this approach accesses the model's internal data paths, it preserves the internal architecture and weights, retaining the advantages of a gray-box setting. It is important to note that while the model layers remain hidden, this setting requires access to their input tokens, and allowing gradients to propagate through them.

### 3.2 ADAPTERS

In this section, we outline a simple solution for the settings discussed above. Our DarkGray-Box Input/Output Adapters (DGA) setting transforms the original model's function $F(x)$ into $BF(Ax)$, where $A$ and $B$ are *lightweight adapters* (linear operators), as opposed to modifying the function $F$ directly. We initialize $A$ and $B$ as the identity function to match $F(x) = BF(Ax)$. The input adapter $A$ learns to transform the model's input into a representation that better aligns with task-specific requirements, while the output adapter $B$ applies a simple linear transformation to the model's output.

Figure 2 provides an overview of our input adapters. Below, we describe the architecture of our input adapters for both text and image modalities, as well as the output adapter applied to the model's output features.

**Visual Input Adapter:** For image inputs, the visual adapter consists of learned 2D convolutional layers that preserve the original dimensions of the input image. Since no activation function is included, the visual adapter functions as an affine transformation on the image pixel space. As we

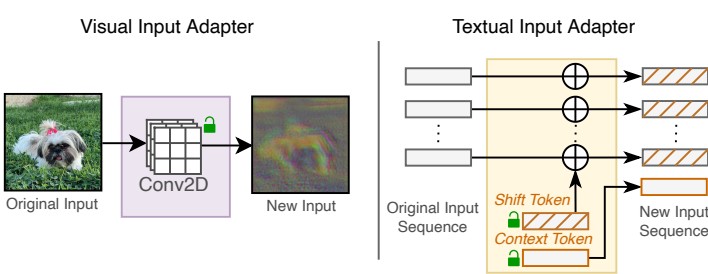

Figure 2: An overview of our Input Adapters. The visual input adapter (left) consists of 2D task-specific convolutional layers that preserve the image's original size. The textual input adapter (right) includes two task-specific tokens: a "shift" token added to the original sequence tokens and an "extra" token appended to the original sequence as a contextual token. Both adapters transform the original input into a new representation that better aligns with the pre-trained backbone model.

observe later (in Section 5), this simplified visual adapter is sufficient for modifying the input for our purposes, and adding non-linear activations does not provide additional benefits.

**Textual Input Adapter:** For text inputs, we draw inspiration from previous works (Li & Liang, 2021; Lester et al., 2021; Liu et al., 2021) and train new textual tokens for the text encoder. However, unlike these methods, we find that optimizing just two tokens—the *extra* token and the *shift* token—is sufficient. The *extra* token is a learned token that is attached to the original input sequence. Due to the transformer's positional invariance (Vaswani et al., 2017), and the fact that positional encoding is not applied to this token, it can be flexibly inserted at any position within the input sequence. The *shift* token is another learned token that is added to each of the original input tokens, effectively "shifting" them within the token embedding space. Thus, this approach requires only one extra token per prompt, which is particularly valuable for text encoders with limited context length (*e.g.* CLIP, which is limited to a total of 77 tokens).

**Intermediate Inputs:** In the LightGray-box setting, we enhance adaptability by injecting learnable, task-specific tokens into each transformer layer of the backbone model $F$. While the model layers remain hidden and fixed, these tokens influence the output of each layer by modifying the attention scores. This approach effectively extends the concept of prompt tuning (Liu et al., 2021) to both visual and textual encoders across multiple tasks.

**Output Adapters:** These adapters are applied to the model's output feature vector. For both image and text modalities, we implement the output adapters as simple linear layer on top of the feature vector space, similar to the linear probing approach (Oquab et al., 2023; Radford et al., 2021).

## 4 EVALUATION

We evaluate DGA and LGA across multiple tasks and benchmarks using various backbones, including CLIP-ViT-B/16, BLIP-B, and DINOv2-B. We compare their performance against the original model in the "Zero-Shot" (ZS) setting as a reference point (serving as a lower bound) and also against the Black-box Linear Probing (LP) baseline. Additionally, we compare them with three strong white-box alternatives that serve as upper bounds: Full Fine-Tuning (FT), Last Layers Fine-Tuning (LLF), and LoRA, as discussed in Sections 1 and 2. Although FT involves the largest number of parameters, it often underperforms compared to lightweight approaches (*e.g.* LoRA, DGA) when the available training samples are insufficient for certain domains or tasks. Since the DGA training paradigm is independent of the backbone architecture, we further demonstrate its performance with CNN backbones in Appendix A. For full implementation details, please refer to Appendix E.

### 4.1 TEXT-TO-IMAGE RETRIEVAL

Table 2 presents a comparison for fine-tuning BLIP on two Text-to-Image Retrieval benchmarks: COCO and Flickr30K. We observe that the FT baseline dominates in both datasets. LoRA, serving

Table 2: Results on two Text-to-Image Retrieval benchmarks, using the BLIP backbone.

| | | | | | | | | |
|---|---|---|---|---|---|---|---|---|
| Full FT | 53.06 | 79.32 | 87.58 | 97.62 | **87.3** | 96.5 | 98.1 | 99.4 |
| Last Layers FT | **54.32** | **80.32** | **87.66** | **97.68** | 86.5 | **96.7** | **98.3** | **99.7** |
| LoRA | 53.48 | 79.78 | 87.46 | 97.6 | 85.4 | 96.6 | 98.1 | 99.6 |
| **LGA (ours)** | 54.14 | 79.72 | 87.48 | 97.66 | 84.7 | 95.9 | 97.7 | 99.4 |
| MaPLe | 52.3 | 78.34 | 86.52 | 97.28 | 84.2 | 96.1 | 97.7 | 99.6 |
| **DGA (ours)** | 53.18 | 79.14 | 87.04 | 97.58 | 83.7 | 95.9 | 97.7 | 99.4 |
| Linear Probing | 51.4 | 78.28 | 86.26 | 97.52 | 83.5 | 95.6 | 97.6 | 99.3 |
| Original (zero-shot) | 47.04 | 74.18 | 83.1 | 96.36 | 78.5 | 94.5 | 96.8 | 98.9 |

as a White-box upper bound, follows closely, while our Gray-box DGA shows a significant improvement with respect to zero-shot, and competitive performance to LoRA, with a recall@1 gap of only 0.38 points on COCO and 2.1 points on Flickr30K. Notably, DGA significantly improves over the ZS baseline, with a recall@1 increase of 5.1 points on COCO and 5.2 points on Flickr30K. LGA slightly improves DGA results by allowing multiple entries to the model's intermediate layers.
To further evaluate DGA and LGA on specific image domains, we created 12 distinct subsets of

Table 3: Performance comparison using the BLIP backbone on different COCO sub-domain splits. Each domain corpus was collected based on human-annotated objects within the images (number of training images in parentheses). Our adapters achieve performance on par with LoRA. The highest values are marked in **bold**, and the second best are underlined.

| | Building (23,021) | | | Furniture (17,882) | | | Grass (22,575) | | | Metal (22,526) | | |
|---|---|---|---|---|---|---|---|---|---|---|---|---|
| | R@1 | R@5 | R@10 | R@1 | R@5 | R@10 | R@1 | R@5 | R@10 | R@1 | R@5 | R@10 |
| Full Fine-tune | 58.47 | 84.54 | 91.18 | 62.51 | **88.59** | 93.48 | 65.2 | 88.76 | 94.97 | 61.8 | 85.2 | 91.53 |
| Last Layers Fine-tune | 60.06 | **85.73** | 91.77 | **63.09** | 87.54 | 93.58 | **68.42** | **91.43** | **95.82** | 62.08 | **86.72** | 91.6 |
| LoRA | 59.66 | 84.74 | **92.07** | 61.84 | 88.3 | 93.48 | 67.02 | 89.94 | 95.61 | **63.18** | 86.51 | **91.95** |
| **LGA (ours)** | **60.26** | 84.14 | 91.48 | 61.94 | 88.11 | **93.67** | 65.42 | 90.58 | 95.61 | 62.15 | 85.78 | 91.67 |
| MaPLe | 58.57 | 83.25 | 91.28 | 60.88 | 86.39 | 92.91 | 63.81 | 89.29 | 94.97 | 61.05 | 85.68 | 91.81 |
| **DGA (ours)** | 58.57 | 83.94 | 91.28 | 61.55 | 87.15 | 91.85 | 65.42 | 90.26 | 95.5 | 61.05 | 85.34 | 91.47 |
| Linear Probing | 56.89 | 83.35 | 90.98 | 60.98 | 86.1 | 92.14 | 64.67 | 89.72 | 94.97 | 59.26 | 84.65 | 90.64 |
| Original (zero-shot) | 52.63 | 80.77 | 87.41 | 56.76 | 83.99 | 90.7 | 59.53 | 87.47 | 93.79 | 56.23 | 81.83 | 89.26 |

| | Paper (9,521) | | | Pavement (18,311) | | | Road (15,402) | | | Sea (6,598) | | |
|---|---|---|---|---|---|---|---|---|---|---|---|---|
| | R@1 | R@5 | R@10 | R@1 | R@5 | R@10 | R@1 | R@5 | R@10 | R@1 | R@5 | R@10 |
| Full Fine-tune | 69.96 | 92.94 | **98.39** | 62.38 | 86.43 | 92.26 | 60.73 | 84.17 | 90.56 | 53.42 | 79.11 | 84.25 |
| Last Layers Fine-tune | 70.16 | **93.15** | 97.38 | **64.29** | 85.71 | **92.74** | **62.25** | 85.08 | 91.02 | **57.53** | 79.11 | **85.62** |
| LoRA | **71.57** | 92.94 | 96.98 | 63.33 | **87.14** | **92.74** | 60.73 | **85.54** | **91.32** | 54.45 | **80.14** | 84.59 |
| **LGA (ours)** | 70.97 | 92.54 | 97.18 | 62.74 | 86.9 | 92.26 | 61.19 | 84.78 | 91.02 | 56.51 | **80.14** | 83.9 |
| MaPLe | 70.16 | 92.54 | 96.37 | 62.38 | 86.19 | 92.38 | 59.97 | 84.02 | 89.95 | 55.48 | 79.11 | 83.9 |
| **DGA (ours)** | 70.56 | 91.73 | 96.57 | 61.55 | 86.9 | 92.14 | 61.04 | 83.56 | 90.56 | 55.82 | 78.42 | 84.59 |
| Linear Probing | 69.76 | 91.33 | 95.97 | 61.43 | 85.0 | 91.55 | 59.51 | 82.65 | 90.26 | 54.45 | 78.08 | 82.19 |
| Original (zero-shot) | 67.74 | 89.92 | 95.56 | 57.62 | 82.98 | 89.4 | 54.49 | 80.37 | 88.13 | 48.29 | 76.37 | 81.16 |

| | Sky (31,808) | | | Table (16,282) | | | Tree (36,466) | | | Window (14,209) | | |
|---|---|---|---|---|---|---|---|---|---|---|---|---|
| | R@1 | R@5 | R@10 | R@1 | R@5 | R@10 | R@1 | R@5 | R@10 | R@1 | R@5 | R@10 |
| Full Fine-tune | 57.06 | 83.91 | 91.46 | 65.3 | 89.18 | **94.99** | 57.05 | 83.74 | 90.23 | 70.91 | 92.87 | 96.53 |
| Last Layers Fine-tune | 59.42 | 84.82 | **91.69** | **66.36** | 90.5 | **94.99** | **59.02** | **85.11** | 91.15 | 71.87 | **93.45** | 96.92 |
| LoRA | 59.27 | **85.58** | 91.53 | 65.7 | 89.45 | 94.72 | 57.7 | 84.2 | **91.21** | **73.8** | **93.83** | 96.92 |
| **LGA (ours)** | **59.73** | 85.13 | 91.38 | 66.23 | 89.84 | 94.33 | 57.9 | 84.79 | 90.69 | 73.41 | 93.26 | **97.11** |
| MaPLe | 57.44 | 84.06 | 91.3 | 65.04 | 88.52 | 94.59 | 56.13 | 83.61 | 90.56 | 70.13 | 93.26 | 96.53 |
| **DGA (ours)** | 58.73 | 84.06 | 90.69 | 65.57 | 89.18 | **94.99** | 56.33 | 83.74 | 90.62 | 71.1 | 92.49 | **97.11** |
| Linear Probing | 56.98 | 83.6 | 90.39 | 62.4 | 87.6 | 94.06 | 55.54 | 83.15 | 90.1 | 68.79 | 92.49 | 96.53 |
| Original (zero-shot) | 52.78 | 80.32 | 87.72 | 59.37 | 83.25 | 92.74 | 51.15 | 79.67 | 88.26 | 67.44 | 91.33 | 95.57 |

the COCO dataset using human annotations to identify objects present in the images. Each subset includes all photos containing a specific element (*e.g.*, table, sky, sea) from both the training and test splits. Table 3 presents the results using the BLIP backbone. Notably, DGA consistently outperforms the ZS and LP baselines across all subsets, demonstrating the effectiveness of modifying the model's inputs and outputs. Interestingly, LoRA outperforms Full Fine-Tuning (FT) in most cases but is itself outperformed by the LLF baseline, highlighting the influence of the number of opti-

mized parameters. Our Gray-box approaches, DGA and LGA, together achieve top-2 performance in 63.89% (23/36) of cases, underscoring their competitive potential.

Table 4: Precision@K comparison on the Stanford-Cars dataset using the BLIP backbone. DGA is competitive with the strongest white-box baseline, Full Fine-Tuning, but both are outperformed by LGA across most metrics.

|                      | P@1   | P@5   | P@10  | P@50  | P@70  |
|----------------------|-------|-------|-------|-------|-------|
| Full Fine-tune       | 98.07 | 98.08 | 97.76 | 77.64 | 57.55 |
| Last Layers Fine-tune| 95.03 | 95.8  | 95.99 | 76.02 | 57.13 |
| LoRa                 | 90.08 | 88.22 | 86.11 | 66.25 | 52.56 |
| **LGA (ours)**       | **98.45** | **98.21** | 97.87 | **77.78** | 57.54 |
| MaPLe                | 97.11 | 97.61 | 97.63 | 77.49 | 57.46 |
| **DGA (ours)**       | 97.16 | 97.91 | **97.97** | 77.53 | **57.59** |
| Linear Probing       | 78.1  | 74.9  | 74.38 | 55.73 | 45.96 |
| Original (zero-shot) | 63.96 | 62.67 | 58.51 | 40.73 | 34.78 |

Next, we conduct an experiment on the domain-specific Stanford-Cars dataset (Krause et al., 2013) as a retrieval task, which contains car images annotated by Make, Model, and Year (*e.g.*, "*2012 Tesla Model S or 2012 BMW M3 Coupe*"). Table 4 presents a Precision@K comparison using the BLIP backbone. Across all metrics, DGA and LGA significantly outperform both the ZS reference and the white-box baselines. Notably, the LoRA baseline underperforms compared to our methods, even though it still shows improvement over the ZS baseline. We attribute this phenomenon to the relatively low number of samples and specific vehicle descriptions (197) in the dataset, making adaptation in the input space more efficient. This suggests that the input adapter's flexibility offers an advantage in such cases. However, this trend is not consistent across all scenarios, as it may vary depending on the backbone model and the dataset used for training.

## 4.2 TEXT-TO-VIDEO RETRIEVAL

Table 5: Performance comparison for fine-tuning BLIP on two Text-to-Video Retrieval benchmarks. Note that due to a small size of training set (7k videos), MSR-VTT full fine-tuning tends to be worse than other methods.

| Model | MSR-VTT | | | | VATEX | | | |
|-------|------|------|------|------|------|------|------|------|
|       | R@1  | R@5  | R@10 | R@50 | R@1  | R@5  | R@10 | R@50 |
| Full Fine-tuning     | 35.96 | 63.96 | 74.28 | 91.33 | **46.97** | **81.13** | **89.17** | **97.97** |
| Last Layers FT       | 36.92 | 64.07 | 74.92 | 91.47 | 44.47 | 78.33 | 87.3  | 97.37 |
| LoRA                 | **37.72** | **65.77** | **76.27** | **92.31** | 41.63 | 75.43 | 84.43 | 96.5  |
| **LGA (ours)**       | 37.04 | 64.14 | 74.29 | 91.36 | 41.23 | 75.43 | 83.6  | 95.67 |
| MaPLe                | 35.17 | 61.33 | 71.9  | 89.79 | 39.03 | 71.53 | 82.27 | 95.37 |
| **DGA (ours)**       | 37.24 | 63.98 | 74.21 | 91.34 | 41.03 | 73.2  | 82.8  | 95.53 |
| Linear Probing       | 35.9  | 62.71 | 72.83 | 90.63 | 38.33 | 70.53 | 80.97 | 94.4  |
| Original (zero-shot) | 32.14 | 56.53 | 66.38 | 85.24 | 31.33 | 61.13 | 71.17 | 89.4  |

Table 5 presents the results of Text-to-Video Retrieval on two benchmarks: MSR-VTT and VATEX. For this task, we follow a previous approach Li et al. (2022) that applies Text-Image foundation models (*e.g.*, CLIP, BLIP) at the frame level for video tasks. Following the established protocol, we uniformly sample 12 frames from each video and perform Text-to-Image Retrieval on the sampled frames. On both benchmarks, DGA achieves results comparable to the LoRA baseline, which performs best on MSR-VTT, with a recall@1 gap of less than one point and a difference of 1 to 2 points at higher recall@k levels. Moreover, DGA significantly outperforms the ZS reference, with a recall@1 improvement of 5.1 points on MSR-VTT and 9.7 points on VATEX. It is notable that the white-box Full Fine-Tuning method outperforms all alternatives on VATEX but surpasses only the zero-shot and linear probing baselines on MSR-VTT. We attribute this to the combination of a high number of trainable parameters and the varying sizes of the training sets, with 26k videos in VATEX compared to only 7k in MSR-VTT. Evidently, the Last Layers Fine-Tuning baseline, with fewer trainable parameters, achieves better results than Full Fine-Tuning on the MSR-VTT dataset.

## 4.3 IMAGE CLASSIFICATION

Table 6: Image Classification results two benchmarks, using CLIP backbone. For ImageNet-1k, we trained with "16-shot" regime, where the training set was limited to 16 random images per class.

| Dataset | Accuracy | Original (ZS) | LP | DGA (ours) | MaPLe | LGA (ours) | LoRA | LL-FT | Full FT |
|---|---|---|---|---|---|---|---|---|---|
| ImageNet1k | Top-1 | 63.87 | 66.91 | 67.77 | 67.49 | 66.94 | 70.29 | 64.11 | 70.79 |
| | Top-5 | 87.82 | 90.23 | 91.66 | 90.83 | 90.45 | 92.81 | 87.31 | 92.29 |
| ImageNet Sketch | Top-1 | 46.97 | 57.30 | 60.06 | 54.57 | 67.48 | 69.04 | 80.97 | 81.05 |
| | Top-5 | 75.23 | 85.56 | 88.27 | 84.17 | 91.12 | 93.73 | 95.12 | 94.96 |

We further evaluate our approach on the Image Classification task using two benchmarks with a CLIP ViT-B/16 backbone, as shown in Table 6. The first classification task on ImageNet 1k (Russakovsky et al., 2014) while the second is sketch-domain classification on ImageNet-Sketch (Wang et al., 2019a). For ImageNet 1k, we perform 16-shot training, sampling 16 images per class from the training set. DGA achieves a 3.9-point improvement in top-1 accuracy over the zero-shot baseline, while on the cross-domain ImageNet-Sketch, it gains a 13.1-point increase. However, LoRA outperforms DGA with a 2.52-point lead on ImageNet-1k and an 8.98-point lead on ImageNet-Sketch. These results suggest that in cross-domain settings, the model requires more substantial internal modifications, which limits the performance of the gray-box approach compared to white-box methods.

## 4.4 SKETCH-TO-IMAGE RETRIEVAL

Table 7: Sketch-to-Image Retrieval results on the Sketchy Dataset, using DinoV2 backbone.

| | All Class | | | | | Novel-Class-25 | | | | |
|---|---|---|---|---|---|---|---|---|---|---|
| | R@1 | R@5 | R@10 | R@50 | R@500 | R@1 | R@5 | R@10 | R@50 | R@500 |
| Full FT | 69.20 | 91.44 | 96.08 | 98.80 | 99.92 | 34.92 | 60.44 | 71.80 | 90.56 | 99.20 |
| Last Layers FT | 65.52 | 91.60 | 95.92 | 98.80 | 100.00 | 30.44 | 56.92 | 68.60 | 90.12 | 99.20 |
| LoRA | 58.72 | 87.44 | 93.76 | 98.88 | 99.92 | 24.76 | 50.40 | 64.36 | 89.20 | 99.32 |
| **LGA (ours)** | 53.36 | 83.84 | 92.32 | 98.56 | 99.92 | 17.88 | 41.72 | 54.28 | 84.64 | 99.32 |
| **DGA (ours)** | 31.20 | 65.92 | 79.76 | 94.48 | 100.00 | 7.44 | 20.16 | 30.28 | 64.60 | 96.16 |
| Linear Probing | 21.12 | 57.92 | 73.84 | 92.24 | 99.20 | 3.72 | 12.80 | 19.44 | 49.08 | 89.92 |
| Original (zero-shot) | 8.56 | 26.64 | 40.16 | 64.08 | 89.28 | 1.80 | 6.76 | 10.64 | 34.80 | 79.08 |

Here we explore Instance Sketch-to-Image Retrieval experiment on the Sketchy dataset (Sangkloy et al., 2016). This dataset includes natural images paired with corresponding human-drawn sketches. The goal is to retrieve *the exact original image* based on a given sketch (not just the class). For this task, we utilized the DinoV2 backbone, which has previously demonstrated strong image feature learning capabilities (Oquab et al., 2023). Notably, this backbone was trained on natural images, resulting in poor performance in the zero-shot setting, as shown in Table 7. Nonetheless, DGA achieves substantial improvement over the zero-shot baseline while keeping the backbone frozen and modifying only the input and output adapters. However, as this task involves adapting to a domain quite different from the original training domain, white-box methods like LoRA, Full FT, and LLF significantly outperform our approach due to their ability to modify model weights. Additionally, LGA, which can adjust internal attention scores, also outperforms DGA and LP by a large margin. These results suggest that for distinct domains, frozen backbones have limitations, and achieving optimal performance requires recalculating more refined internal features.

## 5 ABLATION STUDY

In this section, we explore several key components of DGA and LGA. We start by analyzing the impact of each adapter on the overall performance, the we examine the individual contributions of the adapter's components.

**Impact of Input/Output Adapters**: We start by demonstrating the contribution of each adapter in our DGA approach, both individually and in combination, using the model in zero-shot (ZS) mode as a reference baseline. For each configuration, we train on the COCO (Lin et al., 2014) dataset and report the results on its 5k validation set. Our findings indicate that each input and output (I/O)

Table 8: Ablation study on the COCO 5k validation set, with the CLIP model encoders.

| Baseline | Input Adapter | | Output Adapter | | Recall@K | | | |
| | Vision | Text | Vision | Text | R@1 | R@5 | R@10 | R@50 |
|---|---|---|---|---|---|---|---|---|
| Original (ZS) | ✗ | ✗ | ✗ | ✗ | 31.58 | 55.70 | 66.82 | 89.40 |
| DGA-I-txt | ✗ | ✔ | ✗ | ✗ | 35.78 | 62.02 | 72.90 | 92.70 |
| DGA-I-vis | ✔ | ✗ | ✗ | ✗ | 34.76 | 59.16 | 69.30 | 90.86 |
| DGA-I | ✔ | ✔ | ✗ | ✗ | 37.30 | 63.66 | 74.24 | 93.22 |
| DGA-O-txt | ✗ | ✗ | ✗ | ✔ | 40.76 | 67.72 | 78.18 | 95.18 |
| DGA-O-vis | ✗ | ✗ | ✔ | ✗ | 41.60 | 68.46 | 78.72 | 95.30 |
| DGA-O | ✗ | ✗ | ✔ | ✔ | 41.12 | 69.20 | 79.30 | 95.50 |
| DGA-Text | ✗ | ✔ | ✗ | ✔ | 40.92 | 68.62 | 79.00 | 95.32 |
| DGA-Vis | ✔ | ✗ | ✔ | ✗ | 41.88 | 68.74 | 78.72 | 95.10 |
| DGA | ✔ | ✔ | ✔ | ✔ | **43.04** | **70.52** | **80.26** | **95.94** |

adapter, for both image and text modalities, individually improves the overall performance, as shown in Table 8. In the first three rows, we examine the influence of the input adapters for both modalities (denoted by the "DGA-I" prefix). Each adapter enhances overall performance, and combining both input adapters leads to a 5.72-point gain in Recall@1, demonstrating the effectiveness of modifying the input space of $F$. Adding output adapters to both modalities further improves performance by an additional 5.74 points over the input adapters, forming the complete DGA configuration. Next, we investigate the impact of applying output adapters (denoted by the "DGA-O" prefix) on both the visual and text modalities, which establish a stronger baseline by modifying the output (feature) space of $F$. We then test the mutual influence of both input and output (I/O) adapters in isolation for each modality (shown in the "DGA-Text/Vis" rows). Our findings indicate that combining both I/O adapters for a single modality branch yields better performance than using them separately. Finally, the last row shows the best performance achieved by DGA when all input and output adapters are applied to both the text and image branches. This comprehensive setup consistently outperforms configurations where adapters are applied in isolation or partially, confirming that jointly optimizing all adapters delivers the most significant improvements. These experiments highlight that leveraging both input and output spaces together results in the most effective adaptation of the foundation model $F$ for downstream tasks.

**Textual Input Adapter tokens:** We evaluate the contribution of each learned token in DGA, specifically the *shift* and *extra* tokens, as shown in Table 14. Both tokens improve performance in the lower recall metrics (R@1 and R@5), with the shift token having a minimal effect on higher recall metrics. Additionally, we investigate the impact of using multiple extra tokens, *i.e.* learning more than one input token to be inserted into the prompt, as detailed in Appendix B. Our results indicate that optimizing multiple extra tokens does not consistently outperform using a single token, and it also reduces the effective context length of the encoder (77 tokens in CLIP). Further ablations over the number of proxy tokens in LGA and their layer choices are presented in Appendix B.

## 6 SUMMARY AND DISCUSSION

In this paper, we addressed the challenges of fine-tuning pre-trained foundation models by introducing two novel approaches. The first, the DarkGray-box setting, keeps the model layers and weights concealed, allowing adapters to operate only on the model's input and output. The second, the LightGray-box setting, offers limited access to the model's internal structure, enabling modifications to attention scores without exposing the model's weights. We also discussed the risks of model theft, where it is possible to create a similarly performing model using output features, though this process can be very expensive in terms of time and computational resources. However, if a model's structure and weights are fully exposed, it becomes easy for others to violate intellectual property (IP) rights and use it without incurring any cost or effort, which undermines the original investment. While LGA is tailored for transformer-based architectures, our proposed DGA approach, which employs only input and output adapters, is applicable to a wide range of foundation models, including CNN-based architectures (as demonstrated in Appendix A), such as CLIP and other multimodal models. This generality allows our approach to adapt effectively across various downstream tasks and domains. However, our experiments indicate that this form of adaptation is less effective for more distant domains (*e.g.* sketch), where modifying the model's weights becomes essential. Despite this, our method demonstrates robustness and adaptability, achieving results that are often competitive with, and sometimes surpass, white-box alternatives.

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

# Appendix

In this appendix, we provide comprehensive details about the methods, experiments, and findings discussed in the main paper. We begin with additional evaluations, including experiments on CNN-based backbones, presented in Appendix A. Next, in Appendix B, we perform further ablation studies, exploring the number of input tokens for the input adapter in DGA and the choice of layers in LGA. We also include visualizations of the visual input adapter's operation in Appendix C, offering insights into how it transforms input images. Furthermore, in Appendix D, we delve into an extended discussion on recent advancements in black-box prompt optimization and their limitations, as well as a comparison of task/domain handling schemes using input/output adapters. Finally, Appendix E provides the complete implementation details of our experiments, including training setups, hyperparameters, and model configurations.

## A  FURTHER EVALUATION

Table 9: Evaluating DGA on all CLIP models based on CNN.

| Model # | R@1 | R@5 | R@10 | R@50 |
|---|---|---|---|---|
| CLIP-RN50 - FT | 43.64 | 72.34 | 82.22 | 96.12 |
| CLIP-RN50 - DGA | 32.92 | 60.50 | 72.36 | 92.76 |
| CLIP-RN50 - ZS | 26.46 | 50.30 | 61.58 | 86.88 |
| CLIP-RN101 - FT | 44.90 | 74.16 | 83.40 | 96.66 |
| CLIP-RN101 - DGA | 35.90 | 63.08 | 74.12 | 93.60 |
| CLIP-RN101 - ZS | 27.94 | 52.02 | 63.22 | 87.70 |
| CLIP-RN50X4 - FT | 47.28 | 76.42 | 84.72 | 97.02 |
| CLIP-RN50X4 - DGA | 38.74 | 66.40 | 76.64 | 95.04 |
| CLIP-RN50X4 - ZS | 31.12 | 54.62 | 65.70 | 89.30 |
| CLIP-RN50X16 - FT | 50.48 | 77.50 | 86.04 | 97.44 |
| CLIP-RN50X16 - DGA | 43.18 | 70.34 | 80.54 | 95.98 |
| CLIP-RN50X16 - ZS | 33.98 | 57.78 | 67.86 | 89.46 |

In this section, we further evaluate DGA on the following CLIP CNN-based models: CLIP-RN101, CLIP-RN50, CLIP-RN50x4, and CLIP-RN50x16. Table 9 presents the results on the COCO 5k validation set. Our DarkGray-box approach consistently improves upon the zero-shot (ZS) baseline across all backbones, although it remains inferior to the White-box Full Fine-Tuning (FT) baseline.

We evaluate only these three approaches since these backbones are based on CNN architectures. While it is theoretically possible to apply LoRA to these CNN-based models, it is not straightforward due to the need to carefully select layers and adapt LoRA's implementation to CNN layers. Additionally, LGA is specifically tailored to transformer encoder architectures, making it unsuitable for these CNN backbones.

Table 10 presents a further evaluation of the CLIP backbone on the COCO subsets described in Section 4. We observe similar trends as with the BLIP backbone, where DGA consistently outperforms the ZS and LP baselines. However, white-box methods that have access to model weights continue to outperform DGA and LGA, which leverage a frozen model.

## B  FURTHER ABLATION STUDY

In this section, we present additional ablation studies on the components of DGA and LGA. Table 11 shows the ablation study on the number of input tokens optimized for the text encoder, with BLIP backbone. As observed, the optimal number of tokens lies between 1 and 8. However, it is not entirely clear which number is definitively optimal, as some metrics improve at the expense of others. For example, optimizing 2 tokens yields higher Recall@1 results compared to optimizing 1 token, but results in a lower Recall@5. Nevertheless, the differences across all token numbers

Table 10: Performance comparison using the CLIP backbone on different COCO sub-domain splits. Each domain corpus was collected based on human-annotated objects within the images (number of training images in parentheses). Our adapters achieve performance on par with LoRA.

| | Building (23,021) | | | Furniture (17,882) | | | Grass (22,575) | | | Metal (22,526) | | |
|---|---|---|---|---|---|---|---|---|---|---|---|---|
| | R@1 | R@5 | R@10 | R@1 | R@5 | R@10 | R@1 | R@5 | R@10 | R@1 | R@5 | R@10 |
| Full Fine-tune | 47.18 | 77.6 | 87.22 | 48.9 | 78.81 | 87.92 | 53.75 | 83.4 | 91.54 | 49.35 | 76.46 | 85.89 |
| Last Layers Fine-tune | **54.11** | 80.08 | **89.89** | 56.57 | **83.7** | 90.51 | 58.78 | **87.26** | **94.0** | **56.16** | **82.38** | **90.23** |
| LoRA | 53.32 | **80.77** | 88.4 | **58.2** | 83.51 | **91.28** | **59.21** | 86.08 | **94.0** | 55.61 | 80.87 | 89.06 |
| **LGA (ours)** | 52.43 | 78.79 | 87.41 | 56.95 | 82.36 | 90.12 | 55.46 | 84.9 | 92.72 | 54.3 | 79.49 | 87.68 |
| MaPLe | 49.36 | 76.81 | 85.93 | 55.7 | 80.54 | 89.07 | 53.75 | 83.3 | 92.29 | 53.34 | 78.53 | 86.51 |
| **DGA (ours)** | 49.75 | 75.62 | 83.85 | 52.73 | 79.29 | 88.69 | 52.03 | 81.26 | 89.72 | 49.55 | 76.19 | 84.31 |
| Linear Probing | 46.78 | 73.24 | 83.55 | 52.83 | 78.91 | 87.34 | 52.03 | 80.19 | 89.83 | 48.86 | 75.43 | 84.72 |
| Original (zero-shot) | 35.88 | 61.15 | 71.75 | 44.68 | 70.66 | 79.77 | 40.36 | 67.67 | 80.62 | 40.67 | 65.79 | 75.64 |

| | Paper (9,521) | | | Pavement (18,311) | | | Road (15,402) | | | Sea (6,598) | | |
|---|---|---|---|---|---|---|---|---|---|---|---|---|
| | R@1 | R@5 | R@10 | R@1 | R@5 | R@10 | R@1 | R@5 | R@10 | R@1 | R@5 | R@10 |
| Full Fine-tune | 59.68 | 85.69 | 93.35 | 54.76 | 81.43 | 87.86 | 48.86 | 80.21 | 87.37 | 43.15 | 69.86 | 80.82 |
| Last Layers Fine-tune | **65.12** | **89.11** | **95.77** | 57.98 | 83.1 | 88.93 | 56.77 | 79.91 | **89.19** | 48.97 | **75.34** | **82.19** |
| LoRA | 63.51 | 88.1 | 94.76 | **61.55** | **84.52** | **90.24** | **58.75** | **81.58** | **89.19** | **49.66** | 75.0 | 81.51 |
| **LGA (ours)** | 61.29 | 87.7 | 94.35 | 59.52 | 82.5 | 89.05 | 55.86 | 80.82 | 88.13 | 47.95 | 74.32 | 80.82 |
| MaPLe | 59.27 | 87.9 | 93.75 | 57.98 | 80.0 | 88.69 | 54.34 | 79.0 | 87.52 | 46.92 | 71.58 | 78.42 |
| **DGA (ours)** | 59.07 | 86.29 | 92.94 | 53.33 | 79.4 | 85.71 | 53.58 | 77.17 | 85.39 | 40.75 | 70.55 | 80.82 |
| Linear Probing | 58.87 | 85.48 | 91.94 | 52.38 | 78.93 | 86.07 | 50.84 | 77.02 | 83.71 | 42.81 | 70.89 | 78.42 |
| Original (zero-shot) | 52.62 | 77.42 | 86.69 | 40.95 | 65.95 | 76.31 | 38.51 | 62.71 | 73.36 | 36.3 | 59.93 | 71.23 |

| | Sky (31,808) | | | Table (16,282) | | | Tree (36,466) | | | Window (14,209) | | |
|---|---|---|---|---|---|---|---|---|---|---|---|---|
| | R@1 | R@5 | R@10 | R@1 | R@5 | R@10 | R@1 | R@5 | R@10 | R@1 | R@5 | R@10 |
| Full Fine-tune | 47.44 | 77.12 | 87.87 | 55.15 | 81.53 | 88.65 | 46.43 | 77.9 | 86.36 | 56.07 | 85.16 | 91.33 |
| Last Layers Fine-tune | **52.1** | **81.92** | **90.31** | 58.71 | **85.62** | 91.82 | 53.05 | **81.9** | **89.38** | **65.9** | 88.44 | 94.61 |
| LoRA | 51.33 | 80.32 | 89.24 | **59.63** | 83.25 | **92.35** | **53.11** | 80.52 | 88.79 | 64.93 | **88.63** | **95.38** |
| **LGA (ours)** | 49.43 | 78.49 | 87.87 | 57.39 | 83.77 | 91.42 | 50.49 | 79.87 | 87.08 | 64.35 | 88.05 | 95.18 |
| MaPLe | 48.51 | 77.04 | 87.57 | 58.18 | 82.98 | 89.84 | 47.61 | 77.38 | 86.03 | 63.2 | 87.48 | 94.03 |
| **DGA (ours)** | 45.16 | 75.9 | 85.43 | 54.22 | 81.13 | 88.52 | 47.15 | 75.8 | 84.66 | 59.92 | 86.51 | 93.06 |
| Linear Probing | 43.17 | 73.91 | 84.82 | 53.56 | 80.87 | 88.65 | 45.31 | 74.56 | 83.34 | 59.92 | 86.13 | 93.06 |
| Original (zero-shot) | 34.86 | 61.4 | 73.07 | 44.46 | 72.3 | 80.47 | 35.74 | 61.64 | 73.31 | 50.87 | 80.15 | 87.86 |

Table 11: Ablation study on the number of optimized input tokens, in the text input adapter.

| Tokens # | R@1 | R@5 | R@10 | R@50 |
|---|---|---|---|---|
| **1** | 53.16 | 79.02 | 86.92 | 97.52 |
| **2** | 53.26 | 78.98 | 86.84 | 97.50 |
| **4** | 52.80 | 79.12 | 86.90 | 97.54 |
| **8** | 53.16 | 79.12 | 86.66 | 97.46 |
| **16** | 52.72 | 78.94 | 86.38 | 97.46 |
| **32** | 51.42 | 78.22 | 85.84 | 97.32 |
| **64** | 50.94 | 78.00 | 85.54 | 97.46 |
| **128** | 51.32 | 77.76 | 85.64 | 97.40 |

are minimal, making their performance nearly on par. Consequently, we choose to optimize only 1 token to preserve the text-encoder context length from being occupied by these "proxy" tokens.

Table 12: Ablation study on choice of layers in for the proxy vectors.

| Layers # | R@1 | R@5 | R@10 | R@50 |
|---|---|---|---|---|
| **No FT (zero-shot)** | 42.02 | 69.28 | 79.34 | 95.02 |
| **First layers (0-3)** | 43.10 | 70.16 | 80.08 | 95.80 |
| **Middle layers (4-7)** | 44.56 | 71.22 | 81.20 | 96.16 |
| **Final layers (8-11)** | 44.76 | 71.80 | 81.58 | 96.36 |
| **All layers (0-11)** | 44.88 | 72.56 | 81.98 | 96.26 |

Table 13: Ablation study on the number of learned proxy vector per layer in LGA, on the CLIP backbone.

| Tokens # | R@1 | R@5 | R@10 | R@50 |
|---|---|---|---|---|
| **1** | 44.54 | 71.80 | 81.42 | 96.16 |
| **2** | 44.60 | 72.28 | 81.88 | 96.12 |
| **4** | 45.40 | 72.12 | 81.98 | 96.32 |
| **8** | 45.46 | 72.82 | 82.44 | 96.22 |
| **16** | 46.08 | 73.32 | 82.46 | 96.34 |
| **32** | 46.12 | 73.50 | 82.46 | 96.44 |
| **64** | 46.42 | 73.68 | 82.40 | 96.36 |

Table 14: Ablation study on the textual input adapter components, shift and extra token, on the CLIP backbone.

| Token | R@1 | R@5 | R@10 | R@50 |
|---|---|---|---|---|
| **Only Extra** | 35.32 | 61.28 | 72.08 | 92.14 |
| **Only Shift** | 33.92 | 59.28 | 70.52 | 91.46 |
| **Both** | 35.80 | 61.34 | 72.30 | 92.54 |

**Number of proxy tokens**: In Table 12, we conduct an ablation study on the choice of layers where the proxy vector is learned in LGA. This experiment is carried out on CLIP's visual encoder, trained on the COCO dataset. Injecting proxy vectors into the initial layers of the transformer encoder has a minimal effect, only slightly improving upon the zero-shot baseline, whereas the final layers have the most significant impact. However, using all transformer layers yields the best overall performance, eliminating the need for manual layer selection.

Next, examine the number of learned proxy vectors per layer in our LGA baseline, as presented in Table 13. Generally, increasing the number of learned vectors (and parameters) enhances the model's performance. However, we observe saturation in the Recall@10 and Recall@50 metrics starting from 8 learned vectors. It is important to note that as more vectors are learned, the gradient dimensionality required to propagate through the model to the learned parameters increases, resulting in a trade-off with the amount of information exposed in the Gray-box approach.

Table 15: Ablation study on number of the BLIP last layers fine-tuning, on the COCO dataset.

| Layers # | R@1 | R@5 | R@10 | R@50 |
|---|---|---|---|---|
| **1** | 54.12 | 80.36 | 87.74 | 97.72 |
| **2** | 54.16 | 80.74 | 87.64 | 97.86 |
| **3** | 54.22 | 80.64 | 88.00 | 97.80 |
| **4** | 54.16 | 80.78 | 87.88 | 97.74 |
| **5** | 53.60 | 80.30 | 88.02 | 97.74 |
| **All** | 53.86 | 79.62 | 87.88 | 97.62 |

In Table 15 we ablate over the number of BLIP last layers fine-tuning. Each model was trained on COCO training set, results presented on COCO 5k validation set. We observe minor differences on performance between the methods, where fine-tuning all the layers results in lower performance. We relate it to the high number of parameters versus the low size of training set.

## C   VISUALIZATION

In this section, we visualize the image transformations produced by the input adapter. Figure 3 shows randomly sampled images from the COCO dataset. Each original image is processed through the input adapter and normalized to the same mean and standard deviation as the original image for visualization. Although the transformed images may appear corrupted or unnatural to the human

eye, the model interprets these modified versions more effectively, as evidenced by performance improvements across multiple benchmarks.

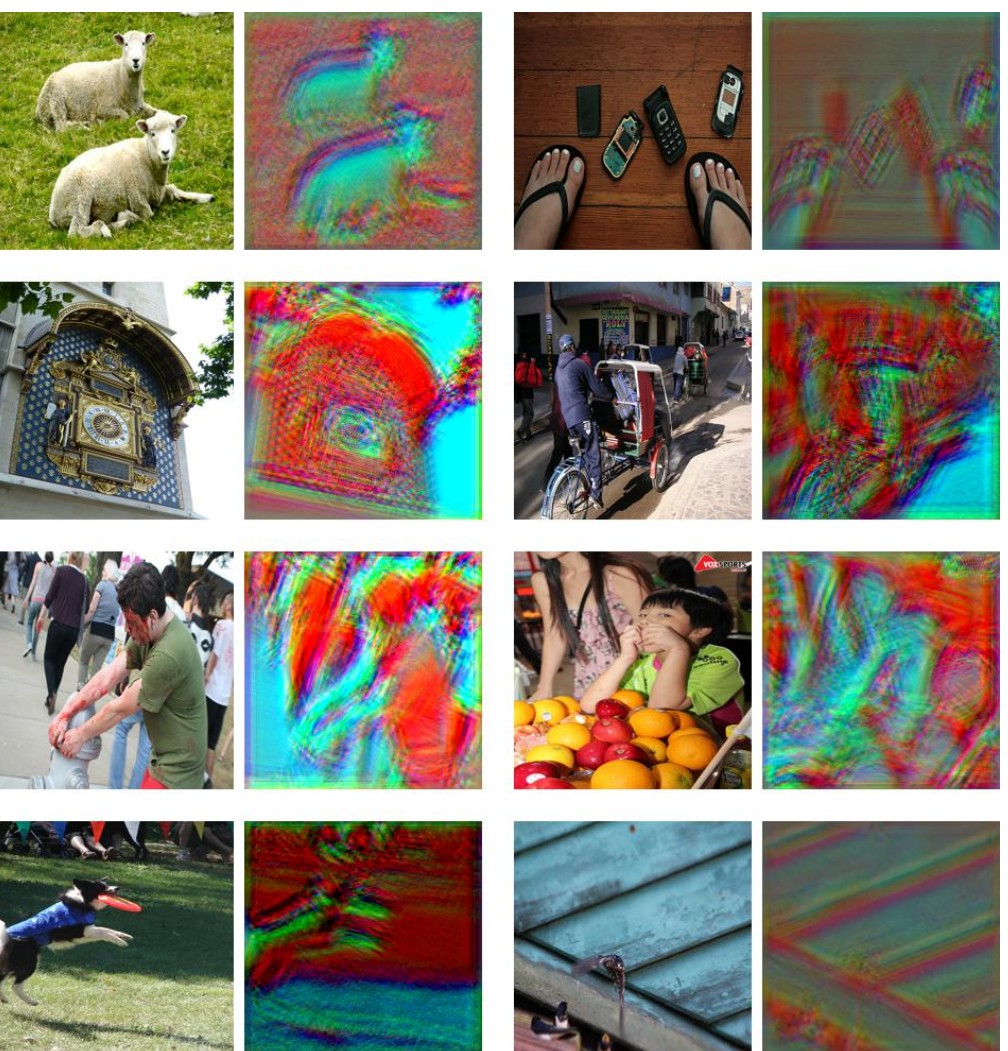

Figure 3: Visualization of the input adapter's influence on images.

## D FURTHER DISCUSSION

Recent studies Liu et al. (2024); Wang et al. (2024) have proposed black-box prompt optimization techniques for Vision-Language models, aiming to enhance performance without requiring access to the backbone model. These methods achieve this by optimizing the input textual prompt, focusing exclusively on text manipulation Wang et al. (2024) or text-to-text mapping Liu et al. (2024), without addressing the visual modality. More specifically, they are designed to optimize textual prompts for tasks such as 16-shot classification. However, this approach limits their applicability to scenarios heavily reliant on the visual domain. For instance, tasks such as Video or Sketch retrieval, which are fundamentally based on visual inputs, remain outside the capabilities of these methods. In contrast, our work addresses such visual domain challenges, expanding the utility and applicability of black-box fine-tuning to a broader range of tasks beyond text-focused optimizations.

To further illustrate the broader applicability of our approach, Figure 4 presents a demonstration of general schemes for handling multiple tasks or domains. The bottom part of the figure illustrates the naive approach of managing each task or domain with its own optimized model. In contrast, the top

part of the figure shows a single optimized backbone model capable of handling all inputs with the use of input/output adapters. First, each input is processed using the appropriate lightweight input adapter. Next, the aggregated batch across all tasks is fed into the model, which produces outputs for each item. Finally, each output is post-processed with its corresponding output adapter to generate the final result.

**Experimental Validation**: To substantiate these claims, we conducted inference experiments comparing two setups: 1) A single backbone combined with 10 pairs of DGA adapters (for 10 different tasks or domains), 2) Ten separate backbones without using our DGA framework. In each setup, we utilize CLIP encoders to encode 10 sampled sets of 100 pairs of images (224x224) and their captions, a total of 1,000 paired samples.

The results demonstrate significant computational and memory efficiency with our approach: Our framework required 22.760 GFLOPs for 1000 samples, compared to 203.223 GFLOPs for the separate backbone setup. Similarly, GPU memory usage was reduced to 1.462 GB, as opposed to 14.54 GB in the alternative setup. These results highlight the resource efficiency and scalability of our framework in managing diverse tasks or domains.

## E  IMPLEMENTATION DETAILS

This section provides the implementation details of our experiments. All methods are trained using the *AdamW* optimizer, with training conducted on 1-4 nodes of *NVIDIA A100* GPUs, depending on the batch size. The input/output adapters are initialized as identity functions.

**Learning Rates**: For CLIP backbones, we train DGA with an initial learning rate of $1 \times 10^{-4}$, and $5 \times 10^{-5}$ for BLIP and DinoV2, all with an exponential decay rate of $0.93$ down to a minimum of $1 \times 10^{-6}$.

**Batch Sizes**: We use a batch size of 256 for all retrieval tasks, except for the Stanford-Cars dataset, where a batch size of 64 is applied. For ImageNet1k classification, a batch size of 1024 is used, and 64 for ImageNet-Sketch.

**Epochs:** We train the models for the following number of epochs on each benchmark: 25 for Stanford-Cars and ImageNet1k (16 shots), 30 for Sketchy and ImageNet-Sketch, 50 for COCO, 2 for Flickr30k, 20 for MSR-VTT, and 40 for VATEX.

**LoRA Hyper-parameters**: For the LoRA baseline, we adapt the $Q$, $K$, and $V$ matrices across all transformer layers, ensuring the rank matches the number of parameters used by DGA and LGA, depending on the backbone.

**Trainable Parameters**: The number of trainable parameters depends on the backbone. For BLIP-B, DGA optimizes 0.10% of the parameters, 0.42% for CLIP, and 1.57% for DINOv2. To ensure a fair comparison, we train the LoRA baselines with a rank $r$ that results in a matched number of trainable parameters to DGA: $r = 8$ for CLIP, $r = 2$ for BLIP, and $r = 25$ for DINOv2. For LGA, we train a proxy token for each of the 12 transformer layers, resulting in a maximum of $12 \cdot 2 \cdot 768$ trainable parameters, depending on the backbone's dimensionality and the number of modalities (image and text).

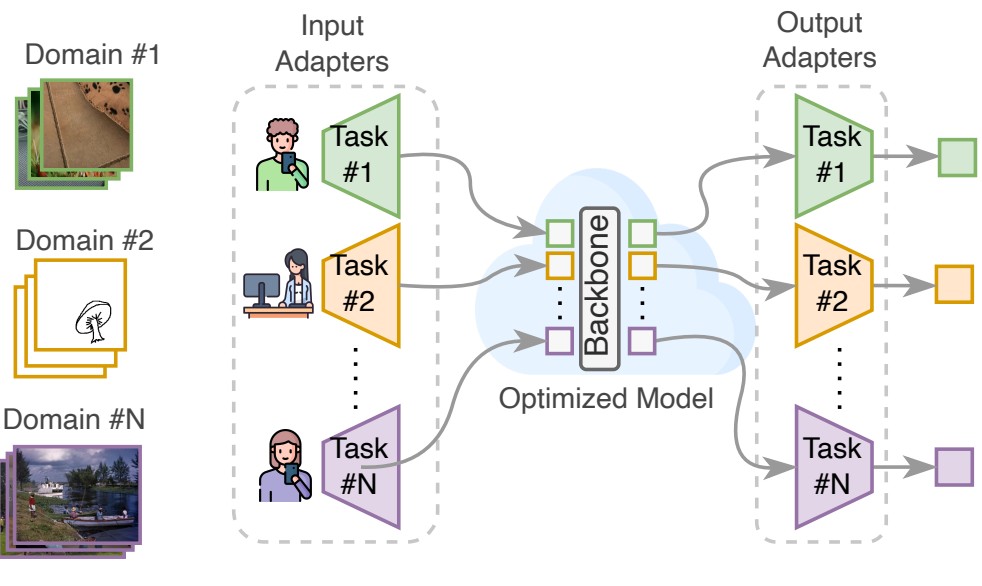

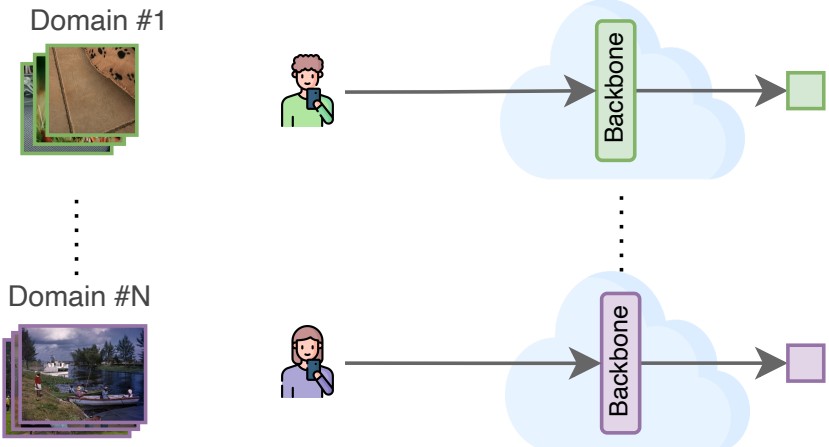

Figure 4: General schemes for handling $N$ different tasks or domains. **Top**: A single optimized model designed for multiple tasks or domains. **Bottom**: A naive approach with $N$ different models, one for each task.

