# OpenReview forum: "Gray-Box Fine-Tuning for Single Backbone Domain Experts"
_ICLR.cc/2025/Conference — Submitted to ICLR 2025_

### Official Review · Reviewer_ZYrt · 2024-11-03

**Soundness:** 3
**Presentation:** 3
**Contribution:** 2
**Rating:** 5
**Confidence:** 3

**Summary:**

The manuscript presents a Gray-box fine-tuning framework designed to balance model adaptation flexibility with privacy and IP protection. In contrast to full fine-tuning, gray-box fine-tuning restricts access to model weights and layers, providing only limited points for gradient propagation, which allows efficient adaptation while keeping core architecture concealed. Two alternatives are presented: DarkGray-box, which confines modifications to input and output layers, and LightGray-box, which exposes additional entry points in intermediate layers for enhanced adaptability. The proposed method is evaluated on several backbones and benchmarks.

**Strengths:**

1. The Gray-box framework introduces a significant advance in balancing model adaptability with privacy, a critical requirement in sensitive applications (e.g., medical data), by enabling model use without exposing proprietary information.
2. Through extensive evaluation on text-image, text-video, and sketch-image benchmarks, the framework shows adaptability across multiple modalities and backbone architectures, illustrating its utility across various domains

**Weaknesses:**

1.	While the paper includes comparisons with prominent methods, additional baselines, such as more recent adapter-based fine-tuning techniques or advanced parameter-efficient methods in federated learning, could provide a more comprehensive view of the framework’s strengths and weaknesses. For example, Liu Shihong et al., in "Language Models as Black-Box Optimizers for Vision-Language Models" (CVPR 2024), Wang Zhengbo et al. Connecting the Dots: Collaborative Fine-tuning for Black-Box Vision-Language Models, (ICML2024), and Lu Wang et al., in "ZooPFL: Exploring Black-box Foundation Models for Personalized Federated Learning" (Arxiv 2310.05143).
2.	In Tab. 8, there is a lack of a comparison of the proposed method with Co-CoOp and MaPLe, which belongs to the gray-box framework.

**Questions:**

Please see the weakness part, the authors should address the following: 1) a discussion clarifying the distinctions between the proposed Gray-box setting and federated learning, as the two share some similarities but also have key differences; and 2) a more extensive baseline comparison. While results for LoRA and Linear Probing are provided on several benchmarks, additional baselines would offer a more comprehensive evaluation.

---

> ### Author Response · Authors · 2024-11-21
>
> **Dear reviewer ZYrt**, we thank you for your thoughtful review and for acknowledging the significance of our work in balancing model adaptability with privacy. We appreciate your feedback, and we address your main points below:
>
> 1. **Distinction Between Our Method and Federated Learning**:
>
> We understand the need to differentiate our Gray-box approach from federated learning (FL), particularly as terms like “gray/white/black boxes” appear in both contexts. While both prioritize privacy, our framework focuses on **model privacy** whereas FL addresses **data privacy**. Our Gray-box framework seals the foundational model’s structure and weights, focusing on secure **task adaptation**.
> Federated learning is a distributed machine learning approach that allows models to be trained on devices without accessing the clients' training data. In this paper, we focus on model privacy rather than data privacy. We added this distinction for clarity, in the revised manuscript.
>
>
> 2. **Additional Baseline Comparisons**:
>
> We have conducted additional comparisons as suggested. Our results demonstrate that our LGA method, operating at the same level of information exposure as MaPLe, consistently outperforms it. These findings have been incorporated into the revised manuscript, with details provided in Tables 3 and 11. Below, we include a few examples of these comparisons for reference.
>
> We sincerely appreciate the reviewer’s suggestion to include comparisons with the cited methodologies. Here, we address the specific works referenced by the reviewer:
>
> 1. **Liu et al., “Language Models as Black-Box Optimizers for Vision-Language Models” (CVPR 2024)**,
>
>    **Wang et al., “Connecting the Dots: Collaborative Fine-Tuning for Black-Box Vision-Language Models” (ICML 2024)**:
>     These papers suggest a black-box prompt optimization for VLLM, and are limited to text manipulation or text-to-text mapping, without altering the other modality. More specifically, they optimize textual prompts, for 16-shot classification tasks. Therefore, they are incapable of performing tasks such as in Video or Sketch retrieval as we demonstrate in our paper, that are based on (or limited to) the visual domain. We further added this discussion to Appendix D.
>
> 2. **Lu et al., “ZooPFL: Exploring Black-Box Foundation Models for Personalized Federated Learning” (arXiv 2023)**:
>
>     While the ZooPFL approach is interesting, their code is not publicly available, which limits our ability to directly evaluate its applicability to our tasks. However, ZooPFL addresses personalization challenges in federated learning with black-box models through distributed training across multiple clients, and it is not fully clear to us how their method could be adapted to our specific context. Additionally, the number of trainable parameters employed in their distributed training setup is significantly higher than the parameter-efficient methods we explore.
>
> Given these differences in scope and methodology, we believe that a direct comparison between our framework and these works falls outside the scope of our study. We hope this explanation clarifies our selection of comparative baselines and highlights the contributions of our work.
>
>
> **Comparison examples on a few benchmarks**:
>
> Our results demonstrate that our LGA, operating at the same level of information exposure as MaPLe, consistently outperforms it across several benchmarks. Notably, even our more restrictive DGA approach, which does not rely on intermediate points for gradient propagation, achieves superior results compared to MaPLe in certain cases.
>
> Due to lack of space, please see results in our comment for **reviewer KBkw** or in our revisited manuscript.
>
> We hope these responses address your concerns and are happy to discuss further.

---

> > ### Author Response · Authors · 2024-11-25
> >
> > Dear Reviewer ZYrt,
> >
> > We sincerely appreciate the time and effort you have dedicated to reviewing our work. We hope you had the opportunity to review our response. If there are any additional concerns or questions, please let us know—we would be more than happy to provide further clarification.
> >
> > Thank you once again for your valuable feedback and consideration.

---

### Official Review · Reviewer_KBkw · 2024-11-03

**Soundness:** 3
**Presentation:** 3
**Contribution:** 2
**Rating:** 6
**Confidence:** 2

**Summary:**

This paper provides a way of fine-tuning models without requiring the exposure of the entire model weights. The authors propose a set of fine-tuning options that trade-off model weight exposure and model performance. They show improvement over black-box optimization methods and close the gap to white-box optimization methods, while also allowing for a tradeoff of model visibility between white and black-box optimization methods.

**Strengths:**

This work has run a large number of experiments thoroughly detailing the strengths and weaknesses of their work. Specifically, while they show they are on average slightly worse than white-box methods, they show improvement over zero-shot and linear probe methods. The notable exception is the  Stanford-Cars dataset using the BLIP backbone, where they outperform all baselines, which the authors attribute to the lower number of training samples.

**Weaknesses:**

This paper mentions other grey-box optimization techniques in the related works but doesn't compare against them.

This paper provides a compromise in terms of accuracy and model visibility in terms of what is revealed to the person fine-tuning the model. As a result, they don't show consistent outperformance over the baselines. For me to improve my review I would like to see comparisons against other gray-box training methods. MaPLe was mentioned in the related works as another work that qualifies as a lightgray box training method and it appears their code is publicly available. If the proposed method is shown to outperform other methods with similar level of model visibility consistently I will improve my rating.

**Questions:**

Last Layers FT is listed in the white-box category for evaluations. Since this baseline wouldnt require the entire model weights, only the activation for a set of final layers, wouldnt this be considered a gray-box method? Last layer fine tuning seems to consistently outperform the proposed method.

---

> ### Author Response · Authors · 2024-11-21
>
> **Dear reviewer KBkw**, thanks for your detailed review and recognition of our thorough experiments. We address your key points below:
>
> 1. **Comparisons with Other Gray-Box Methods**:
>
>     Thank you for your suggestion to compare our methods with MaPLe. Following your request, we conducted that comparison. Below we attach qualitative comparisons on different benchmarks. Our results show that our LGA, which operates at the same information exposure level as MaPLe, consistently outperforms it. These results are now included in the revised manuscript (e.g., Tables 3 and 11). For Co-CoOp, unfortunately, their design is not suitable for the retrieval task (as we describe in lines 193-197, main paper), due to the requirement of both modalities (e.g., text, image) to be encoded jointly rather than separately (e.g., as in CLIP). It requires online computation of the embeddings, conditioned on the query.
>
> 2. **Clarification of Last Layers FT**:
>
>     Although Last Layers FT only exposes the final layer weights, it does involve direct access to model layers, which places it in the white-box category according to our definition. In contrast, our gray-box definition strictly avoids any exposure of model weights while permitting different exit points for gradient propagation (as described in lines 72–75). We have clarified this distinction further in the revised manuscript, with updates highlighted in blue for improved clarity.
>
>
> **Comparison examples on a few benchmarks**:
>
> Our results demonstrate that our LGA, operating at the same level of information exposure as MaPLe, consistently outperforms it across several benchmarks. Notably, even our more restrictive DGA approach, which does not rely on intermediate points for gradient propagation, achieves superior results compared to MaPLe in certain cases.
>
>
> **COCO:**
>
> | Method              	| R@1   | R@5   | R@10  | R@50  |
> |-------------------------|--------|--------|--------|--------|
> | Full FT            	| 53.06  | 79.32  | **87.58**  | 97.62  |
> | Last Layers FT     	| **54.32**  | 80.32  | **87.66** | **97.68**  |
> | LoRA               	| 53.48  | **79.78**  | 87.46  | 97.60  |
> | LGA (ours)           	| **54.14** | **79.72**  | 87.48  | **97.66**  |
> | MaPLe              	| 52.30  | 78.34  | 86.52  | 97.28  |
> | DGA (ours)            | 53.18  | 79.14  | 87.04  | 97.58  |
> | Linear Probing     	| 51.40  | 78.28  | 86.26  | 97.52  |
> | Original (zero-shot)| 47.04  | 74.18  | 83.10  | 96.36  |
>
> **Building:**
>
> | Method              	| R@1   	| R@5   	| R@10  	|
> |-------------------------|-----------|-----------|-----------|
> | Full Fine-tune      	| 58.47 	| 84.54 	| 91.18 	|
> | Last Layers Fine-tune   | __60.06__   | **85.73** | __91.77__   |
> | LoRA               	| 59.66 	| __84.74__   | **92.07** |
> | LGA (ours)            	| **60.26** | 84.14 	| 91.48 	|
> | MaPLe              	| 58.57 	| 83.25 	| 91.28 	|
> | DGA (ours)             	| 58.57 	| 83.94 	| 91.28 	|
> | Linear Probing     	| 56.89 	| 83.35 	| 90.98 	|
> | Original (zero-shot)   | 52.63 	| 80.77 	| 87.41 	|
>
>
> **Road:**
> | Method | R@1   | R@5   | R@10  |
> |-------------------------|-----------|-----------|-----------|
> | Full Fine-tune      	| 60.73 	| 84.17 	| 90.56 	|
> | Last Layers Fine-tune   | **62.25** | __85.08__   | __91.02__   |
> | LoRA               	| 60.73 	| **85.54** | **91.32** |
> | LGA (ours)            	| __61.19__   | 84.78 	| __91.02__   |
> | MaPLe              	| 59.97 	| 84.02 	| 89.95 	|
> | DGA (ours)            	| 61.04 	| 83.56 	| 90.56 	|
> | Linear Probing     	| 59.51 	| 82.65 	| 90.26 	|
> | Original (zero-shot)   | 54.49 	| 80.37 	| 88.13 	|
>
> **MSR-VTT:**
>
> | Method | R@1 | R@5 | R@10 | R@50 |
> | ----- | --- | --- | ---- | ---- |
> | Full Fine-tuning      | 35.96 | 63.96 | 74.28 | 91.33 |
> | Last Layers FT       | 36.92 | 64.07 | **74.92** | **91.47** |
> | LoRA                      | **37.72** | **65.77** | **76.27** | **92.31** |
> | LGA (ours)              | 37.04 | **64.14** | 74.29 | 91.36 |
> | MaPLe                    | 35.17 | 61.33 | 71.9   | 89.79 |
> | DGA (ours)              | **37.24** | 63.98 | 74.21 | 91.34 |
> | Linear Probing.       | 35.9   | 62.71 | 72.83 | 90.63 |
> | Original (zero-shot) | 32.14 | 56.53 | 66.38 | 85.24 |
>
> **Stanford Cars:**
> | Method | P@1 | P@5 | P@10 | P@50 | P@70 |
> | ----- | --- | --- | ---- | ---- | ---- |
> | Full Fine-tune | __98.07__ | __98.08__ | 97.76 | __77.64__ | __57.55__ |
> | Last Layers FT | 95.03 | 95.8 | 95.99 | 76.02 | 57.13 |
> | LoRa | 90.08 | 88.22 | 86.11 | 66.25 | 52.56 |
> | LGA (ours) | **98.45** | **98.21** | __97.87__ | **77.78** | 57.54 |
> | MaPLe | 97.11 | 97.61 | 97.63 | 77.49 | 57.46 |
> | DGA (ours) | 97.16 | 97.91 | **97.97** | 77.53 | **57.59** |
> | Linear Probing | 78.1 | 74.9 | 74.38 | 55.73 | 45.96 |
> | Original (zero-shot) | 63.96 | 62.67 | 58.51 | 40.73 | 34.78 |
>
> We hope these responses address your concerns and are happy to discuss further if needed.
> Thank you for your thoughtful feedback.

---

> > ### Comment · Reviewer_KBkw · 2024-11-22
> >
> > Thank you for responding to my questions/concerns. As a result of clarification and extra experiments I have upgraded my rating from a 5 to a 6.

---

### Official Review · Reviewer_mA3M · 2024-11-03

**Soundness:** 3
**Presentation:** 4
**Contribution:** 3
**Rating:** 6
**Confidence:** 3

**Summary:**

The paper raises the concern for exposure of pretrained model weights and layers. The proposal is to (1) adapt two lightweight adaptors at the model’s input and output, and (2) introduce more entry points for trade-off between model exposure and performance. The model is shown to have generality across various downstream tasks and domains and results close to white-box alternatives.

**Strengths:**

The author(s) tries to formulate the challenge of model exposure and consider the high-level issues by considering practical use cases.
The experiments are well designed to verify generality and the ablation study is detailed and insightful.

**Weaknesses:**

As LoRA is a very strong baseline, the paper needs strong support to indicate that the new method with less access to the original model is a better approach. The claim that full model exposure is risky sounds reasonable but lacks strong evidence - there is no experiment showing that the new method avoids IP violation or saves computation cost.

**Questions:**

Could it be possible to have some evidence supporting the motivation of the paper to reduce the model exposure?

---

> ### Author Response · Authors · 2024-11-21
>
> **Dear reviewer mA3M**,
> Thank you for your positive comments on our work. We appreciate your recognition of the practical focus of our study and its generality across tasks. Below, we address your concerns:
>
> 1. **Evidence Supporting Reduced Model Exposure Clarification of Motivation**:
>
> The motivation for reducing model exposure is grounded in the practical and theoretical risks associated with sharing foundational model weights and architectures. As cited in our manuscript (lines 49–58), previous works highlight how unauthorized access to model internals can lead to several critical issues, such as:
> - Unauthorized use of proprietary models (OpenAI, 2023),
> - Recovery of sensitive training data embedded in model weights (Haim et al., 2022),
> - IP violations, including the reconstruction of original unpublished model weights through adaptations like LoRA  (Horwitz et al., 2024).
>
> While we do not claim that our method is completely invulnerable, it does offer a safeguard that counters the assumptions exploited by these prior works. We further expand on these concerns in lines 204–217, detailing risks related to model theft and misuse.
>
> 2. **Computation Cost Savings**:
>
> Efficiency Analysis: Building on findings from prior research (Pope et al., 2023; Lester et al., 2021), which explore service utilization at scale (as discussed in lines 39–43), we highlight that managing multiple specialized models for different tasks or domains results in inherently lower resource utilization compared to employing a single, versatile model capable of handling diverse tasks. This distinction is particularly significant in large-scale industrial batch operations. To further demonstrate this advantage, we have added a simple illustrative figure (Section D, Fig. 4) that highlights the benefits of maintaining a unified model over managing multiple, separate models.
>
> Specifically, our input/output adapters act as pre/post-processors for task-specific data and can be computed independently on separate machines, enabling improved load balancing and efficient resource utilization.
>
> **Experimental Validation:**
>
> To substantiate these claims, we conducted inference experiments comparing two setups:
> 1. A single backbone combined with 10 pairs of DGA adapters (for 10 different tasks or domains).
> 2. Ten separate backbones without using our DGA framework.
> In each setup, we utilize CLIP encoders to encode 10 sampled sets of 100 pairs of images (224x224) and their captions, resulting in a total of 1,000 paired samples and their corresponding feature vectors.
>
> The results demonstrate significant computational and memory efficiency with our approach:
> - Our framework required **22.760 GFLOPs** for 1000 samples, compared to **203.223 GFLOPs** for the separate backbone setup.
> - Similarly, GPU memory usage was reduced to **1.462 GB**, as opposed to **14.54 GB** in the alternative setup.
>
> These results highlight the resource efficiency and scalability of our framework in managing diverse tasks or domains. We have included this analysis and the corresponding figure in our revised version, as we greatly value your feedback and its contribution to improving the clarity and depth of our paper.
>
> We hope these additions address your concerns and provide further evidence for the broad applicability of our approach. We remain open to any further questions or suggestions you may have.

---

> > ### Comment · Reviewer_mA3M · 2024-11-22
> >
> > After reviewing the other reviews and your feedback, I find the work to be robust, and the authors have effectively defended their study when challenged. They have addressed my previous concerns with convincing responses, supported by citations and additional experiments. Their diligent efforts merit an upgrade in my rating from 5 to 6.

---

### Official Review · Reviewer_Zs3x · 2024-11-04

**Soundness:** 3
**Presentation:** 3
**Contribution:** 2
**Rating:** 5
**Confidence:** 4

**Summary:**

This paper discusses the challenges of traditional fine-tuning methods for foundational models, which require access to model weights and layers, leading to issues like managing multiple model copies, inefficiencies in edge device optimization, and concerns over proprietary rights and privacy. To address these challenges, authors propose "Gray-box" fine-tuning approaches that keep the model's architecture and weights hidden, allowing only gradient propagation. They introduce a framework with two lightweight learnable modules at the model's input and output to adapt to new tasks: DarkGray-Box Input/Output Adapters (DGA) and LighGray-box (LGA). The approaches are evaluated on benchmarks for text-image, text-video, and sketch-image alignment, showing competitive performance compared to full-access fine-tuning methods despite limited model access.

**Strengths:**

+ The paper focuses on an interesting and practial practical problem: conducting effective and efficient model finetuning for foundational models with limitted access to the model weights and layers.

+ The proposed "Gray-box" fine-tuning approaches (LightGray-box and DarkGray-box), that keep the model's architecture and weights hidden, allowing only gradient propagation. This helps address challenges for this problem.

+ Experiments are conducted on several text-image, text-video, and sketch-image alignment benchmarks, which demonstrates the effectiveness of the proposed methods. They achieve competitive results compared with White-box baselines and meanwhile keeping the foundation model sealed.

**Weaknesses:**

- The paper claims that a new paradigm is proposed for effectively re-use pre-trained foundation models. But it is not clear whether the proposed DGA and LGA can be applied commonly used LLM, VLLM, Text-to-Image/video generation foundation models for various downstream tasks.

- Experiments can not well support the above claim as well. The studies are mainly on representation learning tasks like text-image, text-video, and sketch-image alignment, image classification using baseline models e.g. CLIP,  BLIP, DINO. There lacks studies on other kinds of foundation models on various tasks such as VQA, image/video captioning, Text-to-image/video generation etc.

- Technical contributions are limitted and the method design is pretty straightforward.

**Questions:**

Please refer to details of above "Weaknesses" sections. Basically it is not clear to me if the proposed method can be generalized to various foundation models and achieve good performances on different kinds of tasks.

---

> ### Author Response · Authors · 2024-11-21
>
> **Dear reviewer Zs3x**, thanks for your valuable feedback and for recognizing the significance of our work. We appreciate your thorough review and address your main points below:
>
> 1. **Generalization to Various Foundation Models and Tasks**
>
>     We thank the reviewer for this comment. While experiments in the main paper focused on representation learning we strongly believe that our approach generalizes to other backbones and tasks. Following this question we conducted additional experiments on image captioning and general language understanding using different backbones. Specifically, we apply our LGA approach on (1) a **VLLM (BLIP-2, 3.7B)** for Image Captioning and (2) an **LLM (DeBERTa-v3-base)** for GLUE benchmark (on the MRPC dataset). Results can be found in the table below. The results demonstrate that our approach performs effectively in new contexts, reinforcing its applicability to a variety of foundational models and tasks.
>
>
> **Image Captioning, VLLM BLIP2 (3.7B):**
> | Method	| BLEU  | BLEU Precision-1 | Length Ratio | Rouge1 | RougeLsum |
> |-----------|-------|------------------|--------------|--------|-----------|
> | Zero-Shot | 10.09 | 41.31 |    83.38    	| 44.62  | 40.58 |
> | LGA (ours)  	   | **12.56** | 48.38 |  **92.06** | 45.27 |  41.24 |
> |  LoRA       |12.41 	    | **48.91** |  90.23 |**45.36** | **41.39** |
>
> **General Language Understanding Evaluation**, on MRPC dataset with LLM Deberta-v3-base:
>
> | Method  | Zero-Shot | LGA | LoRA | Full FT |
> |-----------|-------|------|--------------|--------|
> | Accuracy | 68.38 | 79.65 | 77.20 | **91.17** |
>
>
> 2.  **Method design is straightforward**
>
>     We acknowledge the straightforward nature of our method, but see this simplicity as a key strength - it ensures ease of implementation, reduces the complexity of integration, and facilitates broader adoption across different applications. However, we want to emphasize that a key aspect of our novelty lies in the context of our gray-box fine-tuning framework, which is specifically designed to address scenarios with restricted access to model internals, balancing performance, security, and adaptability.
>
> We hope these points address your concerns and strengthen the evidence for the broad applicability of our approach. We are open for more questions.

---

> > ### Author Response · Authors · 2024-11-25
> >
> > Dear Reviewer Zs3x,
> >
> > We sincerely appreciate the time and effort you have dedicated to reviewing our work.
> > We hope you had the opportunity to review our response. If there are any additional concerns or questions, please let us know—we would be more than happy to provide further clarification.
> >
> > Thank you once again for your valuable feedback and consideration.

---

### Meta-Review · Area_Chair_mTLA · 2024-12-17

**Metareview:**

The paper introduces a "Gray-box" fine-tuning approach, where the model's architecture and weights remain hidden, and only gradient propagation is allowed. In addition, LightGray-box and DarkGray-box are proposed to show competitive results with whitebox approaches for different foundational models. All reviewers agree that the problem is interesting and the discussions are insightful. However, the generalization of the approach to other tasks is still under-explored (in the rebuttal the authors only added image captioning and language understanding evaluation. Since the method focuses on foundational models, it is crucial to show the comprehensiveness of the approach in more evaluations and therefore I would recommend rejection.

**Additional Comments On Reviewer Discussion:**

Reviewers Zs3x, KBkw, and ZYrt asked about additional evaluations and comparisons against other gray-box fine-tuning baselines. The authors added most of the available baselines, and image captioning and language understanding evaluations. However, it would be more convincing if other types of evaluations (VQA, Image/Video generation, etc, as mentioned by Reviewer Zs3x) could be added to the paper.

Reviewer mA3M asked about LoRA comparison, IP protection, and cost saving. The authors presented concrete GFLOPS and GPU memory usage compared to the baseline setup, to support their argument.

---

### Decision · Program_Chairs · 2025-01-22

Reject